# SAVING FOUNDATION FLOW-MATCHING PRIORS FOR INVERSE PROBLEMS

## ABSTRACT

Foundation flow-matching (FM) models promise a universal prior for solving inverse problems (IPs), yet today they trail behind domain-specific or even untrained priors. *How can we unlock their potential?* We introduce FMPlug, a plug-in framework that redefines how foundation FMs are used in IPs. FMPlug combines an instance-guided, time-dependent warm-start strategy with a sharp Gaussianity regularization, adding problem-specific guidance while preserving the Gaussian structures. This leads to a significant performance boost across image restoration and scientific IPs. Our results point to a path for making foundation FM models practical, reusable priors for IP solving.

## 1 INTRODUCTION

Inverse problems (IPs) are prevalent in many fields, such as medical imaging, remote sensing, and computer vision (Aster et al., 2018; Mohamad-Djafari, 2013). In an IP, the objective is to recover an unknown object $\boldsymbol{x}$ of interest from the relevant measurement $\boldsymbol{y} \approx \mathcal{A}(\boldsymbol{x})$, where the mapping $\mathcal{A}(\cdot)$, called the **forward model**, models the measurement process and the approximation sign $\approx$ accounts for potential modeling errors and measurement noise. Due to insufficient measurement and/or the approximate relationship in $\boldsymbol{y} \approx \mathcal{A}(\boldsymbol{x})$, in practice $\boldsymbol{x}$ is typically not uniquely recoverable from $\boldsymbol{y}$ alone, i.e., ill-posedness. So, to obtain reliable and meaningful solutions for IPs, it is important to incorporate prior knowledge of $\boldsymbol{x}$.

Traditional ideas for solving IPs rely on optimization formulations, often motivated under the Maximum A Posteriori (MAP) estimation principle:

$$\min_{\boldsymbol{x}} \ \ell(\boldsymbol{y}, \mathcal{A}(\boldsymbol{x})) + \Omega(\boldsymbol{x}). \tag{1.1}$$

Here, minimizing the data fitting loss $\ell(\boldsymbol{y}, \mathcal{A}(\boldsymbol{x}))$ encourages $\boldsymbol{y} \approx \mathcal{A}(\boldsymbol{x})$, and the regularization term $\Omega(\boldsymbol{x})$ encodes prior knowledge of ideal solutions to resolve ambiguities and hence mitigate potential ill-posedness. The resulting optimization problems are often solved by gradient-based iterative methods. **Advances in deep learning (DL) have revolutionized IP solving**. Different DL-based approaches to IPs operate with variable levels of data-knowledge tradeoffs. For example, supervised approaches take paired datasets $\{(\boldsymbol{y}_i, \boldsymbol{x}_i)\}_{i=1,\ldots,N}$ and directly learn the inverse mapping $\boldsymbol{y} \mapsto \boldsymbol{x}$, with or without using $\mathcal{A}$ (Ongie et al., 2020; Monga et al., 2021; Zhang et al., 2024); alternatively, data-driven priors learned from object-only datasets $\{\boldsymbol{x}_i\}_{i=1,\ldots,N}$ can be integrated with Eq. (1.1) to form hybrid optimization formulations that effectively combine data-driven priors on $\boldsymbol{x}$ and knowledge about $\boldsymbol{A}$, noise, and other aspects (Oliviero-Durmus et al., 2025; Daras et al., 2024; Wang et al., 2024; 2025); strikingly, untrained DL models themselves can serve as effective plug-in priors for Eq. (1.1), without any extra data (Alkhouri et al., 2025; Wang et al., 2023; Li et al., 2023; Zhuang et al., 2023a;b; Li et al., 2021). Ongie et al. (2020); Monga et al. (2021); Alkhouri et al. (2025); Scarlett et al. (2023); Daras et al. (2024); Oliviero-Durmus et al. (2025) give comprehensive reviews of these DL-based ideas.

In this paper **we focus on solving IPs with deep generative priors (DGPs) pretrained on object-only datasets** Oliviero-Durmus et al. (2025). Compared to supervised approaches that need to construct task-specific paired datasets and perform task-specific training, this approach enjoys great flexibility, as DGPs can be plugged into and reused for different IP problems related to the same family of objects. Among the different DGPs, **we are most interested in those based on the**

**emerging flow-matching (FM) framework (Lipman et al., 2024)**—which is rapidly replacing diffusion models as the backbone of increasingly more state-of-the-art (SOTA) deep generative models in various domains (Black Forest Labs et al, 2025; Patrick Esser et al, 2024; Agarwal, Niket et al, 2025) due to its conceptual simplicity and superior performance.

Several recent works have proposed to solve IPs with pretrained FM priors (Daras et al., 2024). Although promising, most of them are based on **domain-specific** FM priors, e.g., trained on the `FFHQ` dataset for human faces and the `LSUN bedrooms` dataset for bedroom scenes. This limits the practicality of these methods, as domain-specific FM models are not always readily available, e.g., due to data or computing constraints. On the other hand, the emergence of domain-agnostic **foundation** FM models, such as Stable Diffusion 3.0 (or newer versions) (Patrick Esser et al, 2024) and Flux.1 (Black Forest Labs et al, 2025) for images, obsoletes domain-specific developments; Kim et al. (2025); Patel et al. (2024); Ben-Hamu et al. (2024); Martin et al. (2025) propose such ideas. **However, the reported performance from these works based on foundation FM priors clearly lags behind those with domain-specific FM priors, and even behind those with untrained priors**; see Section 2.3. This is not entirely surprising, as foundation priors are considerably weaker than domain-specific priors in terms of constraining the objects.

In this paper, we take the first step to close the performance gap. We focus on IPs where the object $\boldsymbol{x}$ is an image, as foundation FM models for images are widely available and image-related IPs find broad applications. To strengthen the foundation FM priors, we consider two practical settings: (A) **simple-distortion setting**, in which $\boldsymbol{x}$ and $\boldsymbol{y}$ are close (e.g., image restoration); and (B) **few-shot setting**, in which a small number of image instances close to $\boldsymbol{x}$ are provided (e.g., scientific IPs). For both settings, taking the image instance(s) close to $\boldsymbol{x}$ as a guide, we develop a time-dependent warm-start strategy and a sharp Gaussian regularization that together lead to convincing performance gains. In summary, our contributions include: **(1) identifying** the performance gap between foundation FM, domain-specific, and untrained priors for solving IPs (Section 2.3); **(2) proposing** a time-dependent warm-start strategy and a sharp Gaussian regularization that effectively strengthen foundation FM priors (Section 3); and **(3) confirming** the effectiveness of the proposed prior-strengthening method through systematic experimentation (Section 4).

## 2 TECHNICAL BACKGROUND & RELATED WORK

### 2.1 FLOW MATCHING (FM)

Flow Matching (FM) models are an emerging class of deep generative models (Lipman et al., 2024). They learn a continuous flow to transform a prior distribution $p_0(\boldsymbol{z})$ into a target distribution $p_1(\boldsymbol{z})$—in the same spirit of continuous normalizing flow (CNF) (Chen et al., 2018; Grathwohl et al., 2019), where the flow is described by an ordinary differential equation (ODE)

$$d\boldsymbol{z} = \boldsymbol{v}(\boldsymbol{z}, t)\, dt. \tag{2.1}$$

Whereas CNF focuses on the density path induced by the flow and performs maximum likelihood estimation as the learning objective, FM tries to learn a parametrized velocity field $\boldsymbol{v}_{\boldsymbol{\theta}}(\boldsymbol{z}, t)$ to match the one associated with the desired flow. To generate new samples after training, one simply samples $\boldsymbol{z}_0 \sim p_0(\boldsymbol{z})$ and numerically solves the learned ODE induced by $\boldsymbol{v}_{\boldsymbol{\theta}}(\boldsymbol{z}, t)$ from $t = 0$ to $t = 1$, to produce a sample $\boldsymbol{z}_1 \sim p_1(\boldsymbol{z})$.

For tractability, in practice, FM matches the conditional velocity field instead of the unconditional one discussed above: for each training point $\boldsymbol{x}$, a simple conditional probability path $p_t(\boldsymbol{z}_t|\boldsymbol{x})$, e.g., induced by a linear flow $\boldsymbol{z}_t = t\boldsymbol{x} + (1 - t)\boldsymbol{z}_0$, is defined. The model $\boldsymbol{v}_{\theta}(\boldsymbol{z}_t, t)$ is then trained to learn the known vector field of these conditional flows, i.e., $\boldsymbol{u}(\boldsymbol{z}_t, t|\boldsymbol{x})$:

$$\min_{\boldsymbol{\theta}}\ \mathbb{E}_{\boldsymbol{x}, \boldsymbol{z}_0, t} \left\| \boldsymbol{v}_{\theta}(\boldsymbol{z}_t, t) - \boldsymbol{u}(\boldsymbol{z}_t, t|\boldsymbol{x}) \right\|^2. \tag{2.2}$$

Diffusion models (DMs) based on probability flow ODEs can also be interpreted as FMs, although (1) they match the score functions $\nabla_{\boldsymbol{z}} \log p_t(\boldsymbol{z})$ induced by the chosen probability path, not the velocity field as in FM; and (2) they typically work with affine flows for convenience, instead of the simple linear flows often taken in FM practice (Lipman et al., 2024; Song et al., 2021). So, FM can be viewed as a general deep generative framework that covers DMs besides other possibilities.

### 2.2 PRETRAINED FM PRIORS FOR IPS

Recent methods that use pretrained FM priors for solving IPs can be classified into two families, as illustrated in Fig. 1: **(1) The interleaving approach** interleaves the ODE generation steps (i.e., numerical integration steps) with gradient steps toward feasibility (i.e., moving $x$ around to satisfy $y \approx \mathcal{A}(x)$) (Pokle et al., 2023; Kim et al., 2025; Patel et al., 2024; Martin et al., 2025; Erbach et al., 2025). Despite the simplicity and empirical effectiveness on simple IPs, these methods might

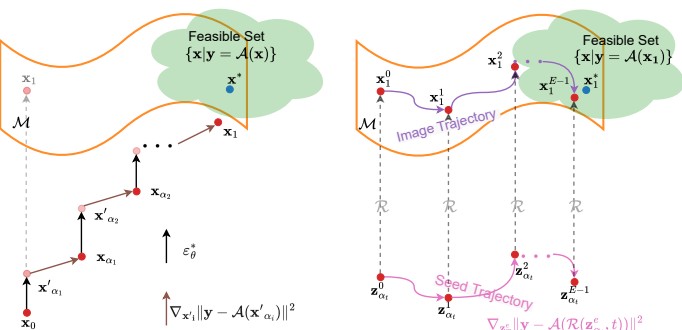

Figure 1: Visual illustration of the difference between the interleaving approach and the plug-in approach to IPs with pretrained FM priors

not converge or return an $x$ that respects the pretrained FM prior (i.e., **manifold feasibility**) or satisfies the measurement constraint $y \approx \mathcal{A}(x)$ (i.e., **measurement feasibility**); and **(2) the plug-in approach** views the generation process as a function $\mathcal{G}_{\theta}$ that maps any source sample to a target sample, and plugs the prior into Eq. (1.1) to obtain a unified formulation (Ben-Hamu et al., 2024):

$$z^* \in \arg\min_z \ \mathcal{L}(z) \doteq \ell(y, \mathcal{A} \circ \mathcal{G}_{\theta}(z)) + \Omega \circ \mathcal{G}_{\theta}(z), \tag{2.3}$$

where $\circ$ denotes functional composition. The estimated object is $\mathcal{G}_{\theta}(z^*)$. Here, the generator $\mathcal{G}_{\theta}$ is fixed and the output $\mathcal{G}_{\theta}(z)$ naturally satisfies the manifold feasibility. In addition, global optimization of $\mathcal{L}(z)$ forces small $\ell(y, \mathcal{A} \circ \mathcal{G}_{\theta}(z))$, and hence $y \approx \mathcal{A} \circ \mathcal{G}_{\theta}(z)$, i.e., leading to measurement feasibility. We note that there is a similar classification of recent work using pretrained diffusion priors to solve IPs; see Wang et al. (2024; 2025); Daras et al. (2024); Oliviero-Durmus et al. (2025).

## 2.3 FOUNDATION FM PRIORS FOR IPS

| | PSNR↑ | SSIM↑ | LPIPS↓ | CLIPIQA↑ |
|---|---|---|---|---|
| **DIP** | 27.5854 | 0.7179 | 0.3898 | 0.2396 |
| **D-Flow (DS)** | **28.1389** | **0.7628** | **0.2783** | **0.5871** |
| **D-Flow (FD)** | 25.0120 | 0.7084 | 0.5335 | 0.3607 |
| **D-Flow (FD-S)** | 25.1453 | 0.6829 | 0.5213 | 0.3228 |
| **FlowDPS (DS)** | 22.1191 | 0.5603 | 0.3850 | 0.5417 |
| **FlowDPS (FD)** | 22.1404 | 0.5930 | 0.5412 | 0.2906 |
| **FlowDPS (FD-S)** | 22.0538 | 0.5920 | 0.5408 | 0.2913 |

Table 1: Comparison between foundation FM, domain-specific FM, and untrained priors for Gaussian deblurring the on AFHQ-Cat dataset (resolution: $256 \times 256$). DS: domain-specific FM; FD: foundation FM; FD-S: strengthened foundation FM; DIP: deep image prior. **Bold**: best, & underline: second best, for each metric/column. The foundation model is Stable Diffusion V3 here.

### 2.3.1 FOUNDATION FM PRIORS $\ll$ DOMAIN-SPECIFIC OR EVEN UNTRAINED ONES

The availability of large-scale training sets has recently fueled intensive development of foundation generative models in several domains, most of them based on FM models and variants, e.g., Stable Diffusion V3 (and newer) (Patrick Esser et al, 2024) and FLUX.1 (Black Forest Labs et al, 2025) for images, OpenAI Sora (OpenAI, 2024) and Google Veo (DeepMind, 2025) for videos, and Nvidia Cosmos world model (Agarwal, Niket et al, 2025). So, recent IP methods based on pretrained FM priors have started to shift from domain-specific priors to these foundation priors.

Although these foundation FM models are powerful enough to generate diverse objects, when used as object priors for IPs, they only constrain the object to be physically meaningful (e.g., the object being a natural image)—**foundation models are powerful as they are not specific**. In comparison, domain-specific priors provide much more semantic and perhaps structural information about the object (e.g., the object being a facial or brain MRI image). So, **foundation priors alone are considerably weaker than domain-specific priors for IPs**. In fact, untrained priors, such as deep image prior (DIP) and implicit neural representation, may be powerful enough to promote physically meaningful solutions for IPs (Alkhouri et al., 2025; Wang et al., 2023; Li et al., 2023; Zhuang et al., 2023a;b; Sitzmann et al., 2020).

A quick comparison summarized in Table 1 confirms our intuition: **recent IP methods with foundation FM priors perform much worse than domain-specific FM, and even untrained, priors** on Gaussian deblurring. Here, Flow-DPS (Kim et al., 2025) and D-Flow (Ben-Hamu et al., 2024) are representative interleaving and plug-in IP methods, respectively. For both of them, foundation priors (`FlowDPS(FD)&D-Flow(FD)`) lag behind domain-specific (`FlowDPS(DS)&D-Flow(DS)`) priors by considerable margins in at least two of the four metrics we report. Moreover, Eq. (1.1) integrated with the untrained DIP is the second best method by three of the four metrics, just after `D-Flow(DS)`. Similarly, results on Gaussian deblurring with varying kernel sizes presented in Fig. 2 show unequivocally that domain-specific and untrained priors are

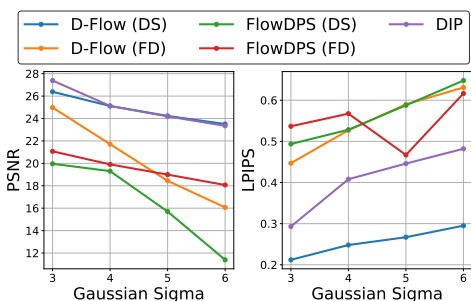

Figure 2: Comparison between foundation FM, domain-specific FM, and untrained priors for Gaussian deblurring with varying kernel size (Gaussian sigma) and hence difficulty level. Notations the same as in Table 1.

stronger than foundation priors, uniformly across different difficulty levels of Gaussian deblurring.

### 2.3.2 CURRENT IDEAS TO STRENGTH FOUNDATION FM PRIORS DO NOT QUITE WORK

While none of the previous works **explicitly** acknowledges and discusses the serious performance issue of foundation FM priors, some have **implicitly** tried to strengthen the priors. As a plug-in method, Ben-Hamu et al. (2024) assumes that $x$ and $y$ are close—e.g., valid for typical image restoration tasks, and initializes the optimization variable $z$ of Eq. (2.3) with

$$z_0 = \sqrt{\alpha}y_0 + \sqrt{1-\alpha}z \quad \text{with } z \sim \mathcal{N}(0, I), \tag{2.4}$$

where $y_0$ is the backward solution of the governing ODE, i.e., $y_0 = y + \int_1^0 v_\theta(y_t, t)dt$, or **inversion seed** in other words, **to accelerate the convergence of numerical methods for solving Eq. (2.3)**. Moreover, they promote the Gaussianity of the seed $z_0$ by recognizing that $\|z_0\|_2^2$ follows a $\chi^2$ distribution and thus regularizes its negative log-likelihood. Alternatively, as a representative interleaving method, Kim et al. (2025) also assumes the closeness of $x$ and $y$, and takes an automatically generated text description for $y$ as text conditions for the FM prior, as all recent foundation FM models allow text-prompted generation. However, **our quick empirical evaluation suggests that these prior-strengthening techniques are almost useless**: there is little change in performance moving from `FlowDPS(FD)&D-Flow(FD)` to `FlowDPS(FD-S)&D-Flow(FD-S)` in Table 1.

## 3 METHOD

The goal of this paper is to close the performance gap between foundation FM priors and domain-specific FM & untrained ones as identified in Section 2.3.1 by addressing the deficiency of current prior-strengthening ideas revealed in Section 2.3.2. We focus on IPs where the object $x$ is an image for our methodology development and validation due to the wide availability of foundation FM models for images, although the proposed method is totally generic and can be easily applied to IPs involving other data modalities as long as relevant foundation FM models are available.

Between the two approaches to solving IPs with pretrained FM priors (Section 2.2), we follow the **plug-in approach** as formulated in Eq. (2.3), due to its superior performance in practice (see, e.g., Table 1 and Section 4). For this approach, a potential concern is whether $\mathcal{G}_\theta$ is surjective, i.e., whether every reasonable $x$ can be represented as $\mathcal{G}_\theta(z)$ for some $z$. While theoretical results of this nature seem lacking and modeling

Table 2: Image regression on 1000 random images from the `DIV2K` dataset; details in Appendix A.2.

| Metric | D-Flow | FMPlug |
|--------|--------|--------|
| PSNR | 36.187 | 37.924 |
| LPIPS | 0.181 | 0.093 |

high-dimensional distributions for such theoretical analysis also seem tricky, empirically, the desired surjectivity seems to hold approximately based on our image regression test reported in Table 2.

To strengthen the foundation FM priors, we consider two practical settings: (A) **simple-distortion setting**, in which $x$ and $y$ are close, e.g., for image restoration. This is the setting considered in previous prior-strengthening works (Ben-Hamu et al., 2024; Kim et al., 2025); and (B) **few-shot**

**setting**, in which a small number of image instances close to $\boldsymbol{x}$ are provided but $\boldsymbol{x}$ and $\boldsymbol{y}$ might not be close. This is particularly relevant for IPs arising from scientific imaging, where the image domain is typically very narrow and is known ahead of time with a few samples (Huang et al., 2022; Shen et al., 2019; Masto et al., 2025). For both settings, taking the image instance(s) close to $\boldsymbol{x}$ as a guide, we develop a time-dependent warm-start strategy and a sharp Gaussian regularization that together lead to convincing performance gains. Below, we first assume the simple-distortion setting and describe the warm-start strategy and the Gaussian regularization in Section 3.1 and Section 3.2, respectively; we then discuss how to extend the ideas to deal with the few-shot setting in Section 3.3.

Gaussianity in the source and intermediate distributions of FM models and especially the following celebrated concentration-of-measure (CoM) result for Gaussian vectors are crucial for our method.

**Theorem 3.1** (Concentration of measure in Gaussian vectors (Vershynin, 2018)). *For a $d$-dimensional $\boldsymbol{z} \sim \mathcal{N}(\boldsymbol{0}, \boldsymbol{I})$, $\mathbb{P}[|\|\boldsymbol{z}\|_2 - \sqrt{d}| \geq t] \leq 2e^{-ct^2}$ for a universal constant $c > 0$.*

This implies that for a standard Gaussian vector $\boldsymbol{z} \in \mathbb{R}^d$, $\|\boldsymbol{z}\|_2$ lies sharply in the range $[(1 - \varepsilon)\sqrt{d}, (1 + \varepsilon)\sqrt{d}]$ with $\varepsilon \in o(1)$ with overwhelmingly high probability. In other words, $\boldsymbol{z}$ lies in an ultra-thin shell around $\mathbb{S}^{d-1}(\boldsymbol{0}, \sqrt{d})$ (a sphere in $\mathbb{R}^d$ centered at $\boldsymbol{0}$ and with a radius $\sqrt{d}$).

### 3.1 AN INSTANCE-GUIDED & TIME-DEPENDENT WARM-START STRATEGY

**Why is the warm-start strategy in D-Flow problematic?** In the standard FM setting, the source distribution $\boldsymbol{z}_0 \sim \mathcal{N}(\boldsymbol{0}, \boldsymbol{I})$, whereas the initialized $\boldsymbol{z}_0$ in Eq. (2.4) has a distribution $\mathcal{N}(\sqrt{\alpha}\boldsymbol{y}_0, (1 - \alpha)\boldsymbol{I})$. One might not worry about this distribution mismatch, as both are supported on the entire ambient space in theory. But finite-sample training in practice causes a significant gap: due to CoM of Gaussian vectors (Theorem 3.1), virtually all training samples drawn from $\mathcal{N}(\boldsymbol{0}, \boldsymbol{I})$ come from an ultra-thin shell $\mathcal{S}$ around $\mathbb{S}^{d-1}(\boldsymbol{0}, \sqrt{d})$, so the generation function $\mathcal{G}_{\boldsymbol{\theta}}$ is effectively trained on inputs from the domain $\mathcal{S}$, not the entire ambient space: the behavior of $\mathcal{G}_{\boldsymbol{\theta}}$ on $\mathcal{S}^c$, the complement of $\mathcal{S}$, is largely undetermined. Now, samples from $\mathcal{N}(\sqrt{\alpha}\boldsymbol{y}_0, (1 - \alpha)\boldsymbol{I})$ concentrate around another ultra-thin shell around $\mathbb{S}^{d-1}(\sqrt{\alpha}\boldsymbol{y}_0, \sqrt{(1 - \alpha)d})$, which has only a negligibly small intersection with $\mathcal{S}$ and lies mostly in $\mathcal{S}^c$. So, the initialization in Eq. (2.4) lies in $\mathcal{S}^c$ with a very high probability. Given that the behavior of $\mathcal{G}_{\boldsymbol{\theta}}$ on $\mathcal{S}^c$ can be wild, this initialization strategy is problematic.

**Our time-dependent warm-up strategy** A typical flow of FM models takes the form

$$\boldsymbol{z}_t = \alpha_t \boldsymbol{x} + \beta_t \boldsymbol{z} \quad \text{where } \boldsymbol{z} \sim \mathcal{N}(\boldsymbol{0}, \boldsymbol{I}), \tag{3.1}$$

where $\alpha_t$ and $\beta_t$ are known functions of $t$. Now, when $\boldsymbol{x}$ and $\boldsymbol{y}$ are close, $\boldsymbol{x} = \boldsymbol{y} + \boldsymbol{\varepsilon}$ for some small $\boldsymbol{\varepsilon}$. So, we can write the flow as

$$\boldsymbol{z}_t = \alpha_t(\boldsymbol{y} + \boldsymbol{\varepsilon}) + \beta_t \boldsymbol{z} \quad \text{where } \boldsymbol{z} \sim \mathcal{N}(\boldsymbol{0}, \boldsymbol{I}) \tag{3.2}$$

for an unknown $\boldsymbol{\varepsilon}$. When $\alpha_t$ is sufficiently small—i.e., we are sufficiently close to $t = 0$ in the flow, $\alpha_t \boldsymbol{\varepsilon}$ can be negligibly small, leading to the approximate flow

$$\boldsymbol{z}_t \approx \alpha_t \boldsymbol{y} + \beta_t \boldsymbol{z} \quad \text{where } \boldsymbol{z} \sim \mathcal{N}(\boldsymbol{0}, \boldsymbol{I}). \tag{3.3}$$

In practice, we do not know how small $\alpha_t$ should be, so we leave it learnable, leading to

$$\boxed{\min_{\boldsymbol{z}, t \in [0,1]} \ \ell(\boldsymbol{y}, \mathcal{A} \circ \mathcal{G}_{\boldsymbol{\theta}}(\alpha_t \boldsymbol{y} + \beta_t \boldsymbol{z}, t))} \tag{3.4}$$

Here we overload the notation of $\mathcal{G}_{\boldsymbol{\theta}}$ as $\mathcal{G}_{\boldsymbol{\theta}} : \mathbb{R}^d \times [0, 1] \to \mathbb{R}^d$—the second input is the current $t$ on the path (the notation in Eq. (2.3) assumes $t = 0$). In other words, due to the closeness of $\boldsymbol{x}$ and $\boldsymbol{y}$, we do not need to start from scratch, i.e., from a random sample drawn from the source distribution; instead, we plug $\boldsymbol{y}$ into an appropriate, learnable time point of the flow to create a shortcut.

Our formulation in Eq. (3.4) can be easily generalized to latent FM models that are commonly used in practice—we just need to replace $\mathcal{A} \circ \mathcal{G}_{\boldsymbol{\theta}}$ with $\mathcal{A} \circ \mathcal{D} \circ \mathcal{G}_{\boldsymbol{\theta}}$ for the decoder $\mathcal{D}$ in use. Moreover, it is not only grounded in theory and effective in practice (see Section 4), but also speeds up learning as $t > 0$ implies shorter flows, although improving speed is not our current focus.

**Additional mean-variance calibration**   Due to approximation errors in matching the ideal flow during FM training, as well as when approximating Eq. (3.2) using Eq. (3.3), the distribution of $z_t$ could be slightly off the ideal distribution. To rectify this, we perform a scalar mean-variance calibration in our implementation: we first draw 4000 unconditional samples from the foundation FM model and estimate the scalar mean and variance of all coordinates for each time step on the FM model's time grid; we then fit the data using a lightweight neural network, which predicts mean and variance as a continuous function of $t \in [0, 1]$, to be compatible with our continuous optimization in Eq. (3.4). Our mean-variance calibration follows

$$\widehat{z}_t = \sqrt{\sigma^2(Z_t)/\sigma^2(z_t)} \cdot (z_t - \mu(z_t)) + \mu(Z_t), \tag{3.5}$$

where $\mu(Z_t)$ and $\sigma^2(Z_t)$ are the scalar mean and variance predicted by the neural network, and $\mu(z_t)$ and $\sigma^2(z_t)$ are scalar mean and variance for $z_t$ across all coordinates.

## 3.2   A SHARP GAUSSIANITY REGULARIZATION

**Why is the Gaussian regularization in D-Flow problematic?**   If $z_0 \sim \mathcal{N}(0, I)$, $\|z_0\|_2^2 \sim \chi^2(d)$ and the negative log-likelihood is $h(z_0) = -(d/2-1)\log\|z_0\|_2^2 + \|z_0\|_2^2/2 + C$ for some constant $C$ independent of $z_0$. Ben-Hamu et al. (2024) promotes the Gaussianity of $z_0$ by regularizing $h(z_0)$. While $h(z_0)$ is minimized at any $z_0$ satisfies $\|z_0\|_2 = \sqrt{d-2}$, away from this value the function changes painfully slowly; see Fig. 3. For example, the function value only changes $\leq 0.031\%$ relative to the

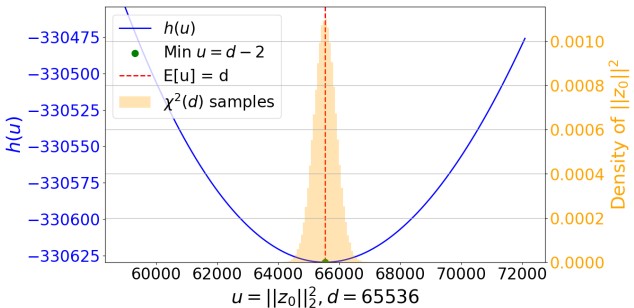

Figure 3: Plot of the function $h(z_0)$ (after a change of variable $u = \|z_0\|_2^2$). An ideal regularization function should blow up sharply away from the narrow concentration region in orange to promote Gaussianity effectively.

minimum in the $[62000, 70000]$ range, much larger than the orange-highlighted CoM region. This is problematic, as $\|z_0\|_2$ should concentrate sharply around $d$ and thus only functions that blow up quickly away from the $\|z_0\|_2 = \sqrt{d}$ level can effectively promote the Gaussianity of $z_0$.

**Our sharp Gaussian regularization via an explicit constraint**   For Eq. (3.4), we hope to promote the Gaussianity of $z$. To enforce the sharp concentration of $z$, we introduce the shell constraint

$$(1 - \varepsilon)\sqrt{d} \leq \|z\|_2 \leq (1 + \varepsilon)\sqrt{d}, \quad \text{with an } \varepsilon \ll 1 \tag{3.6}$$

as implied by Theorem 3.1. To ensure feasibility, in each iteration step to optimize Eq. (3.4), we simply need to add the closed-form projection

$$z' = \begin{cases} (1 + \varepsilon)\sqrt{d} \cdot z/\|z\|_2 & \text{if } \|z\|_2 \geq (1 + \varepsilon)\sqrt{d} \\ (1 - \varepsilon)\sqrt{d} \cdot z/\|z\|_2 & \text{if } \|z\|_2 \leq (1 - \varepsilon)\sqrt{d} \\ z & \text{otherwise} \end{cases} \tag{3.7}$$

Using a spherical constraint $\|z\|_2 = \sqrt{d}$ or regularization to promote Gaussianity is not new in the FM and diffusion literature; see, e.g., Yang et al. (2024). However, enforcing $\|z\|_2 = \sqrt{d}$ is a bit rigid as the actual length lies in a small range. Our shell constraint leaves reasonable slackness while still sharply encoding the Gaussianity. We typically set $\varepsilon = 0.025$ in our implementation.

## 3.3   EXTENSION INTO THE FEW-SHOT SETTING

We assume a small set of instances $\{x_k\}_{k=1,\dots,K}$, all of which are close to the true $x$. To adapt the time-dependent warm-start strategy in Section 3.1 to this setting, we consider linear combinations of $x_k$'s to take the place of $y$ for warm-start, i.e., starting with $\alpha_t(\sum_{k=1}^{K} w_k x_k) + \beta_t z$, resulting in

$$\min_{z, t \in [0,1], w \in \Delta^K} \ell(y, \mathcal{A} \circ \mathcal{G}_\theta(\alpha_t(\textstyle\sum_{k=1}^{K} w_k x_k) + \beta_t z, t)) \tag{3.8}$$

to replace Eq. (3.4), where the simplex constraint $\Delta^K \doteq \left\{ \boldsymbol{w} \in \mathbb{R}^K : \boldsymbol{w} \geq \boldsymbol{0}, \boldsymbol{w}^\intercal \boldsymbol{1} = 1 \right\}$ fixes the scale of $\boldsymbol{w}$, as the multiplicative relationship of $\alpha_t$ and $\boldsymbol{w}$ causes scale ambiguity. In actual implementation, we eliminate this constraint by a simple reparametrization $\boldsymbol{w} = \mathrm{softmax}(\boldsymbol{v})$ and treat $\boldsymbol{v}$ as a group of optimization variable. Since the proposed modification in warm-start does not affect $\boldsymbol{z}$, our sharp Gaussian regularization in Section 3.2 can be directly integrated.

# 4 EXPERIMENT

Table 3: Results on simple-distortion IPs. (**Bold**: best, under: second best, CLIP: CLIPIQA)

| task | method | AFHQ ($512 \times 512$) | | | | DIV2K ($512 \times 512$) | | | | RealSR ($512 \times 512$) | | | |
|---|---|---|---|---|---|---|---|---|---|---|---|---|---|
| | | PSNR ↑ | SSIM ↑ | LPIPS ↓ | CLIP ↑ | PSNR ↑ | SSIM ↑ | LPIPS ↓ | CLIP ↑ | PSNR ↑ | SSIM ↑ | LPIPS ↓ | CLIP ↑ |
| Super Resolution ×4 | DIP | 29.85 | 0.78 | 0.37 | 0.33 | 25.75 | 0.73 | 0.42 | 0.40 | **26.81** | 0.72 | 0.44 | 0.30 |
| | FlowChef-P | 29.23 | 0.79 | 0.38 | 0.64 | 25.08 | 0.71 | 0.43 | **0.60** | 25.89 | 0.71 | 0.43 | **0.44** |
| | FlowChef | 29.25 | 0.79 | 0.38 | **0.65** | 25.09 | 0.71 | 0.43 | **0.60** | 25.92 | 0.71 | 0.43 | **0.44** |
| | FlowDPS-P | 28.75 | 0.76 | 0.37 | 0.37 | 24.92 | 0.69 | 0.42 | 0.51 | 26.11 | 0.71 | 0.43 | 0.34 |
| | FlowDPS | 28.60 | 0.75 | 0.42 | 0.35 | 24.83 | 0.68 | 0.45 | 0.46 | 26.10 | 0.70 | 0.45 | 0.32 |
| | DFlow | 26.37 | 0.70 | 0.54 | 0.31 | 23.42 | 0.64 | 0.52 | 0.37 | 23.60 | 0.62 | 0.53 | 0.28 |
| | **FMPlug-W** | 30.13 | **0.81** | 0.34 | 0.18 | 25.77 | **0.74** | **0.38** | 0.24 | 26.58 | 0.73 | 0.39 | 0.16 |
| | **FMPlug** | **30.31** | **0.81** | **0.33** | 0.20 | **25.88** | **0.74** | **0.38** | 0.27 | 26.66 | **0.74** | **0.38** | 0.17 |
| Random Inpainting 70% | DIP | **33.32** | **0.90** | **0.21** | 0.47 | 28.49 | **0.86** | **0.27** | 0.59 | 30.88 | **0.89** | **0.25** | 0.47 |
| | FlowChef-P | 29.27 | 0.77 | 0.41 | 0.57 | 24.67 | 0.67 | 0.46 | 0.50 | 25.81 | 0.69 | 0.45 | 0.35 |
| | FlowChef | 29.35 | 0.77 | 0.41 | 0.58 | 24.76 | 0.67 | 0.46 | 0.50 | 25.89 | 0.69 | 0.45 | 0.35 |
| | FlowDPS-P | 27.63 | 0.73 | 0.41 | 0.43 | 24.01 | 0.65 | 0.47 | 0.54 | 25.68 | 0.69 | 0.47 | 0.36 |
| | FlowDPS | 27.53 | 0.72 | 0.47 | 0.35 | 24.04 | 0.64 | 0.50 | 0.47 | 25.78 | 0.69 | 0.48 | 0.32 |
| | DFlow | 28.43 | 0.76 | 0.41 | 0.65 | 24.71 | 0.73 | 0.41 | 0.67 | 25.27 | 0.69 | 0.42 | **0.59** |
| | **FMPlug-W** | 32.75 | 0.88 | 0.37 | 0.63 | 28.82 | 0.85 | 0.33 | 0.68 | 31.30 | 0.88 | 0.28 | 0.56 |
| | **FMPlug** | 32.81 | 0.87 | 0.34 | **0.66** | **28.95** | 0.84 | 0.32 | **0.69** | **31.79** | 0.89 | 0.26 | 0.56 |
| Gaussian Deblur | DIP | 29.39 | 0.77 | **0.39** | 0.30 | 25.23 | 0.70 | 0.43 | **0.38** | 26.17 | 0.70 | 0.46 | 0.28 |
| | FlowChef-P | 23.84 | 0.63 | 0.54 | 0.28 | 20.41 | 0.49 | 0.62 | 0.23 | 21.42 | 0.51 | 0.63 | 0.19 |
| | FlowChef | 23.87 | 0.63 | 0.54 | 0.28 | 20.41 | 0.49 | 0.62 | 0.23 | 21.42 | 0.51 | 0.63 | 0.19 |
| | FlowDPS-P | 24.15 | 0.60 | 0.49 | 0.23 | 20.23 | 0.46 | 0.58 | 0.32 | 21.21 | 0.47 | 0.59 | 0.22 |
| | FlowDPS | 23.69 | 0.58 | 0.55 | 0.15 | 20.22 | 0.45 | 0.61 | 0.20 | 21.21 | 0.47 | 0.61 | 0.17 |
| | DFlow | 25.90 | 0.66 | 0.54 | **0.34** | 23.64 | 0.64 | 0.52 | 0.37 | 23.65 | 0.60 | 0.54 | **0.30** |
| | **FMPlug-W** | 30.38 | **0.79** | 0.40 | 0.22 | 26.05 | 0.72 | 0.43 | 0.29 | 27.05 | 0.72 | 0.44 | 0.21 |
| | **FMPlug** | **30.41** | **0.79** | **0.39** | 0.21 | **26.26** | **0.73** | 0.41 | 0.28 | **27.22** | **0.73** | 0.43 | 0.19 |
| Motion Deblur | DIP | 28.69 | 0.75 | 0.38 | 0.26 | 24.75 | 0.68 | 0.45 | 0.35 | 26.17 | 0.70 | 0.46 | 0.28 |
| | FlowChef-P | 24.77 | 0.66 | 0.50 | **0.37** | 21.27 | 0.54 | 0.57 | 0.34 | 22.50 | 0.56 | 0.56 | 0.26 |
| | FlowChef | 24.78 | 0.66 | 0.50 | **0.37** | 21.28 | 0.54 | 0.57 | 0.34 | 22.51 | 0.56 | 0.56 | 0.26 |
| | FlowDPS-P | 24.81 | 0.64 | 0.46 | 0.28 | 21.07 | 0.51 | 0.54 | 0.39 | 22.50 | 0.55 | 0.55 | 0.27 |
| | FlowDPS | 24.49 | 0.62 | 0.52 | 0.20 | 21.05 | 0.50 | 0.58 | 0.26 | 22.55 | 0.54 | 0.56 | 0.22 |
| | DFlow | 27.81 | 0.73 | 0.48 | 0.35 | 25.21 | 0.70 | 0.47 | **0.42** | 25.86 | 0.69 | 0.47 | **0.31** |
| | **FMPlug-W** | 30.10 | 0.79 | 0.39 | 0.26 | 26.83 | 0.74 | 0.40 | 0.36 | 28.01 | 0.76 | 0.40 | 0.28 |
| | **FMPlug** | **30.43** | **0.81** | 0.37 | 0.28 | **27.38** | **0.78** | **0.36** | **0.42** | **28.63** | **0.79** | 0.37 | 0.30 |

For brevity, we term our method **FMPlug** and benchmark its performance on both simple-distortion and few-shot IPs, in Section 4.1 and Section 4.2, respectively. In Section 4.3, we perform an ablation study to dissect the contributions of the two algorithmic components.

## 4.1 SIMPLE-DISTORTION IPS

**Datasets, tasks, and evaluation metrics** We use 3 diverse datasets: DIV2K (Agustsson & Timofte, 2017), RealSR (Cai et al., 2019) and AFHQ (Choi et al., 2020), 100 random images each, taken from their dataset. We set the image resolution to $512 \times 512$ by resizing and cropping the original. We consider **four linear IPs**: i) $4\times$ super-resolution from $128 \times 128$ to $512 \times 512$; ii) 70% random-mask inpainting; iii) Gaussian deblurring with a kernel size of 61 and standard deviation of 3.0; iv) Motion deblurring with a kernel size of 61 and intensity of 0.5. We add Gaussian noise $\sigma = 0.03$ to all measurements. For metrics, we use PSNR for pixel-level difference, SSIM and DISTS for structure and texture similarity, LPIPS for perceptual difference, and CLIPIQA & MUSIQ for no-reference quality metric.

**Competing methods** We compare our FMPlug (**-W**: warm-up only, Number of Function Evaluations (NFE) = 3) with deep image prior (DIP) (Ulyanov et al., 2020) (an untrained image prior) + Eq. (2.3), D-Flow (NFE = 6) (Ben-Hamu et al., 2024) (a SOTA plug-in method), FlowDPS (NFE = 28) (Kim et al., 2025) (a SOTA interleaving method) and FlowChef (NFE = 28) (Patel et al., 2024) (another SOTA interleaving method). For a fair comparison, we use Stable Diffusion V3 (Patrick Esser et al, 2024) as the backbone for all methods that require foundation priors. We also compare with OT-ODE (Pokle et al., 2023), PnP-Flow (Martin et al., 2025) based on a domain-specific FM model, FHQ-Cat from (Martin et al., 2025). For methods that integrate text prompts, including FlowDPS and FlowChef, we compare two variants with the prompts on and off, respectively; we use postfix **-P** to indicate the prompt-enabled variants. We use the pretrained degradation-aware prompt extractor of Wu et al. (2024) to generate label-style text prompts. We set the CFG scale to 2.0 when text prompts are on. Details of the hyperparameter can be found in Appendix A.1.

Table 3 summarizes the quantitative results; details and visualizations can be found in Appendix A.4. We can observe that: **(1)** Our FMPlug is the overall winner by all metrics but CLIPIQA and MUSIQ, the no-reference metrics, beating the untrained DIP—a strong baseline. FlowChef and FlowDPS, with and without text prompt, lag behind even the untrained DIP by large margins and generate visually blurry and oversmooth images as shown in Fig. 4, highlighting the general struggle of interleaving methods to ensure simultaneous measurement and manifold feasibility; **(2)** For plug-in methods, our FMPlug improves upon D-Flow—our main competitor, by considerable margins based on all metrics but CLIPIQA, showing the solid advantage of our warm-start strategy and Gaussian regularization over theirs; and **(3)** FMPlug further improves PSNR and SSIM slightly over FMPlug-W, with the largest improvement seen in CLIPIQA, showing stronger visual quality. This confirms the benefits brought about by the sharp Gaussianity regularization in Eq. (3.7).

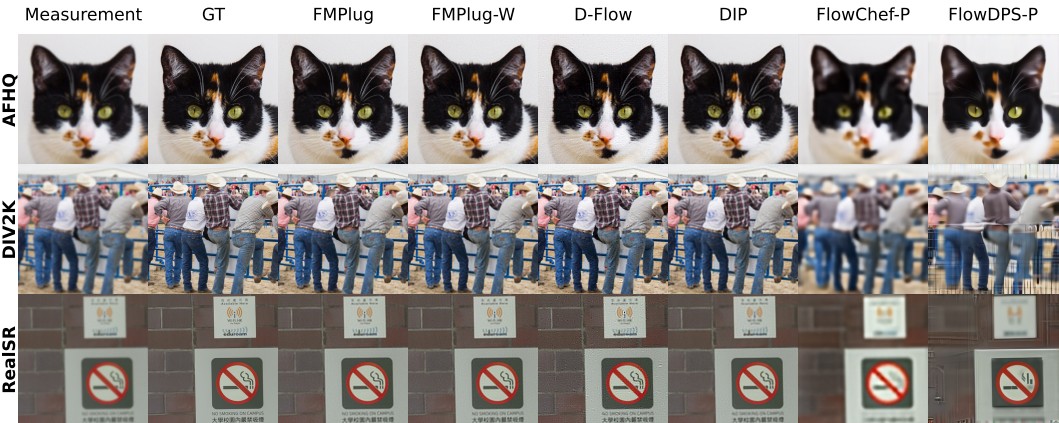

Figure 4: Visual comparison of results in Gaussian deblurring.

Table 4: **Gaussian Deblur** and **Super Resolution** $4\times$ on AFHQ-Cat $256 \times 256$ with additive Gaussian noise ($\sigma = 0.03$). FD: Foundation; DS: Domain-specific; **Bold**: best, under: second best

| | Super Resolution $4\times$ | | | | | | Gaussian Blur | | | | | |
|---|---|---|---|---|---|---|---|---|---|---|---|---|
| | LPIPS↓ | PSNR↑ | SSIM↑ | DIST↓ | CLIPIQA↑ | MUSIQ↑ | LPIPS↓ | PSNR↑ | SSIM↑ | DIST↓ | CLIPIQA↑ | MUSIQ↑ |
| DIP | 0.36 | 28.17 | 0.76 | 0.21 | 0.25 | 28.12 | 0.36 | 27.92 | 0.75 | 0.23 | 0.26 | 23.94 |
| OT-ODE (DS) | **0.19** | 26.43 | 0.74 | 0.90 | 0.59 | **64.63** | **0.19** | 27.67 | 0.75 | 0.89 | 0.62 | **63.82** |
| PnP-Flow (DS) | 0.24 | 27.45 | **0.80** | 0.82 | 0.52 | 51.95 | 0.31 | 28.70 | **0.79** | 0.77 | **0.66** | 40.26 |
| FlowDPS (DS) | 0.24 | 28.56 | 0.79 | **0.14** | 0.57 | 55.63 | 0.38 | 22.27 | 0.56 | **0.20** | 0.52 | 52.42 |
| FlowDPS (FD | 0.37 | 24.45 | 0.74 | 0.27 | **0.63** | 27.96 | 0.55 | 22.11 | 0.59 | 0.38 | 0.28 | 15.10 |
| D-Flow (DS) | 0.27 | 25.81 | 0.69 | 0.82 | 0.52 | 57.74 | 0.20 | 28.41 | 0.77 | 0.87 | 0.61 | 59.29 |
| D-Flow (FD) | 0.53 | 24.64 | 0.67 | 0.31 | 0.31 | 45.27 | 0.56 | 24.42 | 0.62 | 0.21 | 0.30 | 49.12 |
| **FMPlug(FD)** | 0.33 | **28.85** | **0.80** | 0.22 | 0.31 | 28.77 | 0.35 | **29.00** | 0.79 | 0.23 | 0.24 | 30.58 |

To benchmark our progress in bridging the performance gap between foundation and domain-specific priors, we expand Table 1 to include more competing methods and our method into Table 4.

On both Gaussian deblurring and super-resolution, by most of the metrics, our FMPlug gets closer or even comparable to the performance of SOTA methods with domain-specific priors.

## 4.2 FEW-SHOT SCIENTIFIC IPS

We consider two scientific IPs from InverseBench (Zheng et al., 2025) and take their data as necessary: **(1) linear inverse scattering (LIS)**, an IP in optical microscopy, where the objective is to recover the unknown permittivity contrast $z \in \mathbb{R}^n$ from measurements of the scattered light field $y_{sc} \in \mathbb{C}^m$. We use 100 samples for evaluation and 10 samples as few-shot instances; **(2) Compressed sensing MRI**, an important technique to accelerate MRI scanning through subsampling. We use 94 samples from the test set for evaluation and 6 samples from the validation set as instances of a few shots. More details on the forward models can be found in Appendix A.3 and Zheng et al. (2025). For D-Flow, we choose the best result between random initialization and warm-start with the least-loss few-shot instance, trying to make a fair comparison with them.

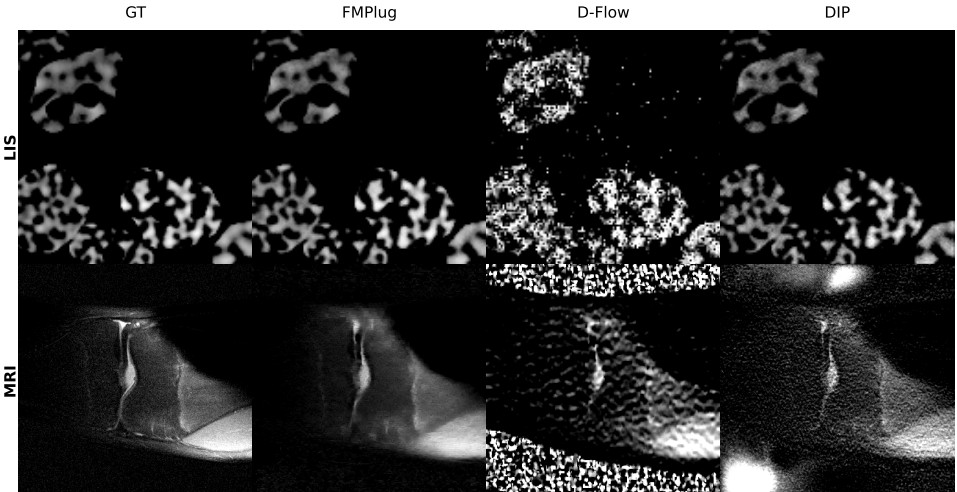

Figure 5: Qualitative comparison of results on knee MRI and LIS. GT: groundtruth

From Table 5, it is evident that in both scientific IPs, our proposed few-shot FMPlug beats both DIP and D-Flow by large margins in PSNR and SSIM. We put Red-Diff, the best SOTA method with domain-specific priors as evaluated in Zheng et al. (2025), as a reference (performance quoted from their paper also), highlighting the gaps to be bridged next. Qualitatively, from Fig. 5, our method faithfully recovers the main object structures, while D-Flow and DIP show severe artifacts.

Table 5: (Scientific IPs) Performance on LIS and MRI. (**Bold**: best among non-DS priors; Background: with DS model)

|  | **LIS** | | **MRI** $(4\times)$ | |
|---|---|---|---|---|
|  | PSNR↑ | SSIM↑ | PSNR↑ | SSIM↑ |
| DIP | 28.72 | 0.96 | 18.35 | 0.39 |
| D-Flow | 17.15 | 0.66 | 8.94 | 0.15 |
| **FMPlug** | **31.83** | **0.97** | **22.94** | **0.48** |
| Red-diff | 36.55 | 0.98 | 28.71 | 0.62 |

## 4.3 ABLATION STUDY

Table 6 shows the performance of FMPlug, and of two variants: FMPlug-Plain (without warm-start and regularization) and FMPlug-W (with warm-start only). Although both ingredients are necessary for the final performance, most of the performance gain comes from the proposed warm-up strategy. The sharp Gaussianity regularization further refines the results.

Table 6: Ablation study on **Gaussian Deblur** on DIV2K with additive Gaussian noise ($\sigma = 0.03$). (**Bold**: best, under: second best). **-W**: with warm-start only

|  | PSNR↑ | SSIM↑ | LPIPS↓ | DIST↓ |
|---|---|---|---|---|
| FMPlug-Plain | 25.1602 | 0.6732 | 0.4846 | 0.1719 |
| FMPlug-W | 26.0547 | 0.7193 | 0.4315 | 0.1620 |
| FMPlug | **26.2563** | **0.7339** | **0.4120** | **0.1565** |

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

# A  APPENDIX

## A.1  EXPERIMENT DETAILS

In this section, we provide implementation details on all methods compared in the experiment section. By default, we use Stable Diffusion V3 Medium[1] (Patrick Esser et al, 2024) as the backbone model whenever foundation FM models are needed.

- **FMPlug** We use `AdamW` as our default optimizer. The number of function evaluations (NFE) is 3 and we use the `Heun2` ODE solver to balance efficiency and accuracy. The learning rate for $z$ is 0.5, and for $t$ is 0.005.
- **D-Flow** We use their default optimizer: `LBFGS` algorithm with line search. The NFE $= 6$ with the `Heun2` ODE solver. We set the weight of their regularization term $\lambda = 0.01$. We perform the initialization with the Euler ODE solver with guidance scale 0.2.
- **FlowDPS** We set NFE $= 28$ with `FlowMatchEulerDiscreteScheduler`. For their data consistency term, we perform it with 3 steps of gradient descent with `step size = 15.0`
- **FlowChef** we set NFE $= 28$ with `FlowMatchEulerDiscreteScheduler`. We use `step size = 50.0` for simple-distortion tasks.
- **Deep Image Prior** We use a 5-layer UNet with 256 channels for each layer with Adam optimizer. We set the learning rate for the network to 0.001.

## A.2  DETAILS ABOUT THE IMAGE REGRESSION EXPERIMENT IN TABLE 2

In the image regression task, we solve

$$z^* \in \arg\min_{z} \ \mathcal{L}(z) \doteq \ell(y, \mathcal{G}_{\theta}(z)) + \Omega \circ \mathcal{G}_{\theta}(z), \tag{A.1}$$

i.e., the forward model $\mathcal{A}$ is the identity map. We use 1000 randomly drawn images from the training set of `DIV2K` and adopt all default hyperparameter settings from Appendix A.1. For D-Flow, we stop optimizing when there is no effective update to $z$ for 5 consecutive epochs. We run FMPlug-W for a maximum of 1000 epochs and use the output as the regression result.

---

[1] https://huggingface.co/stabilityai/stable-diffusion-3-medium

## A.3 Details of Scientific IPs

**Linear inverse scattering (LIS)**  Inverse scattering is an IP in optical microscopy, where the objective is to recover the unknown permittivity contrast $z \in \mathbb{R}^n$ from measurements of the scattered light field $y_{\mathrm{sc}} \in \mathbb{C}^m$. We follow the formulation in  Zheng et al. (2025):

$$y_{\mathrm{sc}} = H(u_{\mathrm{tot}} \odot z) + n \in \mathbb{C}^m \quad \text{where} \quad u_{\mathrm{tot}} = G(u_{\mathrm{in}} \odot z). \tag{A.2}$$

Here, $G \in \mathbb{C}^{n \times n}$ and $H \in \mathbb{C}^{m \times n}$ denote the discretized Green's functions that characterize the optical system response, $u_{\mathrm{in}}$ and $u_{\mathrm{tot}}$ are the incident and total lightfields, $\odot$ represents the elementwise (Hadamard) product, and $n$ accounts for measurement noise.

The resolution of the LIS data is $(1, 128, 128)$. However, Stable Diffusion V3 (SD3) outputs at a resolution $(3, 512, 512)$. So, we downsample the model output in spatial directions to match the spatial dimension of the LIS data. To match the channel dimension, we replicate the single-channel LIS data three times. For evaluation, one of the replicated channels is used as the output.

**Compressed sensing MRI (MRI)**  Compressed sensing MRI (CS-MRI) is an important technique to accelerate MRI scanning via subsampling.  We follow Zheng et al. (2025), and consider the parallel imaging (PI) setup of CS-MRI. The PI CS-MRI can be formulated as an IP in recovering the image $x \in \mathbb{C}^n$:

$$y_j = \mathcal{P}\mathcal{F}S_j x + n_j \quad \text{for } j = 1, ..., J \tag{A.3}$$

where $\mathcal{P} \in \{0, 1\}^{m \times n}$ is the sub-sampling operator and $\mathcal{F}$ is Fourier transform and $y_j$, $S_j$, and $n_j$ are the measurements, sensitivity map, and noise of the $j$-th coil.

The resolution of the MRI images is $(2, 320, 320)$. To deal with the dimension discrepancy with the SD3 output, we again perform spatial downsampling to match the spatial dimensions, and fill in the third channel by the average of the two existing channels. For evaluation, we noly consider the two original channels.

## A.4 Complete results for Table 3

Table 7: **Inpainting** and **Super Resolution** $4\times$ on AFHQ with additive Gaussian noise ($\sigma = 0.03$). (**Bold**: best, under: second best)

| method | Inpainting | | | | | | Super Resolution $4\times$ | | | | | |
|---|---|---|---|---|---|---|---|---|---|---|---|---|
| | PSNR ↑ | SSIM ↑ | LPIPS ↓ | DISTS ↓ | CLIPIQA ↑ | MUSIQ ↑ | PSNR ↑ | SSIM ↑ | LPIPS ↓ | DISTS ↓ | CLIPIQA ↑ | MUSIQ ↑ |
| DIP | **33.32** | **0.90** | **0.21** | 0.07 | 0.47 | 57.73 | 29.85 | 0.78 | 0.37 | **0.12** | 0.33 | 43.38 |
| FlowChef-P | 29.27 | 0.77 | 0.41 | 0.21 | 0.57 | 36.48 | 29.23 | 0.79 | 0.38 | 0.19 | 0.64 | 38.77 |
| FlowChef | 29.35 | 0.77 | 0.41 | 0.21 | 0.58 | 37.02 | 29.25 | 0.79 | 0.38 | 0.19 | **0.65** | 39.01 |
| FlowDPS-P | 27.63 | 0.73 | 0.41 | 0.17 | 0.43 | 56.70 | 28.75 | 0.76 | 0.37 | 0.15 | 0.37 | 52.74 |
| FlowDPS | 27.53 | 0.72 | 0.47 | 0.18 | 0.35 | 49.14 | 28.60 | 0.75 | 0.42 | 0.16 | 0.35 | 47.61 |
| DFlow | 28.43 | 0.76 | 0.41 | 0.17 | 0.65 | 60.45 | 26.37 | 0.70 | 0.54 | 0.18 | 0.31 | **53.13** |
| **FMPlug-W** | 32.75 | 0.88 | 0.37 | 0.08 | 0.63 | 60.87 | **30.13** | **0.81** | 0.34 | 0.13 | 0.18 | 47.43 |
| **FMPlug** | 32.81 | 0.87 | 0.34 | **0.06** | **0.66** | **61.86** | **30.31** | **0.81** | **0.33** | **0.12** | 0.20 | 46.91 |

Table 8: **Gaussian Blur** and **Motion Blur** on AFHQ with additive Gaussian noise ($\sigma = 0.03$). (**Bold**: best, under: second best)

| method | Gaussian Blur | | | | | | Motion Blur | | | | | |
|---|---|---|---|---|---|---|---|---|---|---|---|---|
| | PSNR ↑ | SSIM ↑ | LPIPS ↓ | DISTS ↓ | CLIPIQA ↑ | MUSIQ ↑ | PSNR ↑ | SSIM ↑ | LPIPS ↓ | DISTS ↓ | CLIPIQA ↑ | MUSIQ ↑ |
| DIP | 29.39 | 0.77 | **0.39** | 0.14 | 0.30 | 36.07 | 28.69 | 0.75 | 0.38 | 0.16 | 0.26 | 34.88 |
| FlowChef-P | 23.84 | 0.63 | 0.54 | 0.30 | 0.28 | 15.81 | 24.77 | 0.66 | 0.50 | 0.28 | **0.37** | 19.99 |
| FlowChef | 23.87 | 0.63 | 0.54 | 0.30 | 0.28 | 15.89 | 24.78 | 0.66 | 0.50 | 0.28 | **0.37** | 19.86 |
| FlowDPS-P | 24.15 | 0.60 | 0.49 | 0.24 | 0.23 | 42.74 | 24.81 | 0.64 | 0.46 | 0.21 | 0.28 | 47.77 |
| FlowDPS | 23.69 | 0.58 | 0.55 | 0.27 | 0.15 | 30.28 | 24.49 | 0.62 | 0.52 | 0.24 | 0.20 | 36.63 |
| DFlow | 25.90 | 0.66 | 0.54 | 0.20 | **0.34** | **50.61** | 27.81 | 0.73 | 0.48 | 0.17 | 0.35 | 47.74 |
| **FMPlug-W** | 30.38 | **0.79** | 0.40 | **0.12** | 0.22 | 42.02 | 30.10 | 0.79 | 0.39 | 0.12 | 0.26 | 48.62 |
| **FMPlug** | **30.41** | **0.79** | **0.39** | **0.12** | 0.21 | 43.08 | **30.43** | **0.81** | **0.37** | **0.11** | 0.28 | **52.23** |

Table 9: **Inpainting** and **Super Resolution** 4× on DIV2K with additive Gaussian noise ($\sigma = 0.03$). (**Bold**: best, under: second best)

| method | Inpainting | | | | | | Super Resolution 4× | | | | | |
|---|---|---|---|---|---|---|---|---|---|---|---|---|
| | PSNR ↑ | SSIM ↑ | LPIPS ↓ | DISTS ↓ | CLIPIQA ↑ | MUSIQ ↑ | PSNR ↑ | SSIM ↑ | LPIPS ↓ | DISTS ↓ | CLIPIQA ↑ | MUSIQ ↑ |
| DIP | 28.49 | **0.86** | **0.27** | 0.09 | 0.59 | 55.82 | 25.75 | 0.73 | 0.42 | **0.15** | 0.40 | 37.85 |
| FlowChef-P | 24.67 | 0.67 | 0.46 | 0.24 | 0.50 | 38.04 | 25.08 | 0.71 | 0.43 | 0.22 | **0.60** | 38.50 |
| FlowChef | 24.76 | 0.67 | 0.46 | 0.24 | 0.50 | 38.87 | 25.09 | 0.71 | 0.43 | 0.22 | **0.60** | 38.67 |
| FlowDPS-P | 24.01 | 0.65 | 0.47 | 0.19 | 0.54 | 49.49 | 24.92 | 0.69 | 0.42 | 0.17 | 0.51 | 47.19 |
| FlowDPS | 24.04 | 0.64 | 0.50 | 0.19 | 0.47 | 46.89 | 24.83 | 0.68 | 0.45 | 0.17 | 0.46 | 44.80 |
| DFlow | 24.71 | 0.73 | 0.41 | 0.18 | 0.67 | 62.25 | 23.42 | 0.64 | 0.52 | 0.17 | 0.37 | **57.18** |
| **FMPlug-W** | 28.82 | 0.85 | 0.33 | 0.08 | 0.68 | **65.09** | 25.77 | **0.74** | **0.38** | **0.15** | 0.24 | 40.96 |
| **FMPlug** | **28.95** | 0.84 | 0.32 | **0.07** | **0.69** | 64.80 | **25.88** | **0.74** | **0.38** | **0.15** | 0.27 | 40.30 |

Table 10: **Gaussian Blur** and **Motion Blur** on DIV2K with additive Gaussian noise ($\sigma = 0.03$). (**Bold**: best, under: second best)

| method | Gaussian Blur | | | | | | Motion Blur | | | | | |
|---|---|---|---|---|---|---|---|---|---|---|---|---|
| | PSNR ↑ | SSIM ↑ | LPIPS ↓ | DISTS ↓ | CLIPIQA ↑ | MUSIQ ↑ | PSNR ↑ | SSIM ↑ | LPIPS ↓ | DISTS ↓ | CLIPIQA ↑ | MUSIQ ↑ |
| DIP | 25.23 | 0.70 | 0.43 | 0.18 | **0.38** | 32.54 | 24.75 | 0.68 | 0.45 | 0.20 | 0.35 | 32.59 |
| FlowChef-P | 20.41 | 0.49 | 0.62 | 0.34 | 0.23 | 16.68 | 21.27 | 0.54 | 0.57 | 0.32 | 0.34 | 19.76 |
| FlowChef | 20.41 | 0.49 | 0.62 | 0.34 | 0.23 | 16.68 | 21.28 | 0.54 | 0.57 | 0.32 | 0.34 | 19.82 |
| FlowDPS-P | 20.23 | 0.46 | 0.58 | 0.29 | 0.32 | 35.90 | 21.07 | 0.51 | 0.54 | 0.26 | 0.39 | 39.56 |
| FlowDPS | 20.22 | 0.45 | 0.61 | 0.30 | 0.20 | 30.51 | 21.05 | 0.50 | 0.58 | 0.27 | 0.26 | 34.21 |
| DFlow | 23.64 | 0.64 | 0.52 | 0.17 | 0.37 | **53.03** | 25.21 | 0.70 | 0.47 | 0.17 | **0.42** | **53.78** |
| **FMPlug-W** | 26.05 | 0.72 | 0.43 | **0.16** | 0.29 | 36.66 | 26.83 | 0.74 | 0.40 | 0.14 | 0.36 | 46.95 |
| **FMPlug** | **26.26** | **0.73** | **0.41** | **0.16** | 0.28 | 38.14 | **27.38** | **0.78** | **0.36** | **0.12** | **0.42** | 51.71 |

Table 11: **Inpainting** and **Super Resolution** 4× on RealSR with additive Gaussian noise ($\sigma = 0.03$). (**Bold**: best, under: second best)

| method | Inpainting | | | | | | Super Resolution 4× | | | | | |
|---|---|---|---|---|---|---|---|---|---|---|---|---|
| | PSNR ↑ | SSIM ↑ | LPIPS ↓ | DISTS ↓ | CLIPIQA ↑ | MUSIQ ↑ | PSNR ↑ | SSIM ↑ | LPIPS ↓ | DISTS ↓ | CLIPIQA ↑ | MUSIQ ↑ |
| DIP | 30.88 | **0.89** | **0.25** | 0.09 | 0.47 | 54.97 | **26.81** | 0.72 | 0.44 | **0.17** | 0.30 | 38.23 |
| FlowChef-P | 25.81 | 0.69 | 0.45 | 0.25 | 0.35 | 35.96 | 25.89 | 0.71 | 0.43 | 0.24 | **0.44** | 35.42 |
| FlowChef | 25.89 | 0.69 | 0.45 | 0.25 | 0.35 | 36.61 | 25.92 | 0.71 | 0.43 | 0.23 | **0.44** | 35.65 |
| FlowDPS-P | 25.68 | 0.69 | 0.47 | 0.20 | 0.36 | 49.28 | 26.11 | 0.71 | 0.43 | 0.18 | 0.34 | 46.24 |
| FlowDPS | 25.78 | 0.69 | 0.48 | 0.19 | 0.32 | 46.54 | 26.10 | 0.70 | 0.45 | 0.18 | 0.32 | 44.49 |
| DFlow | 25.27 | 0.69 | 0.42 | 0.21 | **0.59** | 60.99 | 23.60 | 0.62 | 0.53 | 0.20 | 0.28 | **56.53** |
| **FMPlug-W** | 31.30 | 0.88 | 0.28 | 0.07 | 0.56 | **62.77** | 26.58 | 0.73 | 0.39 | **0.17** | 0.16 | 40.05 |
| **FMPlug** | **31.79** | **0.89** | 0.26 | **0.06** | 0.56 | 62.61 | 26.66 | **0.74** | **0.38** | **0.17** | 0.17 | 39.27 |

Table 12: **Gaussian Blur** and **Motion Blur** on RealSR with additive Gaussian noise ($\sigma = 0.03$). (**Bold**: best, under: second best)

| method | Gaussian Blur | | | | | | Motion Blur | | | | | |
|---|---|---|---|---|---|---|---|---|---|---|---|---|
| | PSNR ↑ | SSIM ↑ | LPIPS ↓ | DISTS ↓ | CLIPIQA ↑ | MUSIQ ↑ | PSNR ↑ | SSIM ↑ | LPIPS ↓ | DISTS ↓ | CLIPIQA ↑ | MUSIQ ↑ |
| DIP | 26.17 | 0.70 | 0.46 | 0.20 | 0.28 | 31.78 | 26.17 | 0.70 | 0.46 | 0.22 | 0.28 | 33.25 |
| FlowChef-P | 21.42 | 0.51 | 0.63 | 0.36 | 0.19 | 16.65 | 22.50 | 0.56 | 0.56 | 0.33 | 0.26 | 20.77 |
| FlowChef | 21.42 | 0.51 | 0.63 | 0.36 | 0.19 | 16.68 | 22.51 | 0.56 | 0.56 | 0.33 | 0.26 | 20.88 |
| FlowDPS-P | 21.21 | 0.47 | 0.59 | 0.30 | 0.22 | 38.23 | 22.50 | 0.55 | 0.55 | 0.27 | 0.27 | 41.01 |
| FlowDPS | 21.21 | 0.47 | 0.61 | 0.31 | 0.17 | 33.68 | 22.55 | 0.54 | 0.56 | 0.28 | 0.22 | 37.84 |
| DFlow | 23.65 | 0.60 | 0.54 | 0.20 | 0.30 | **54.62** | 25.86 | 0.69 | 0.47 | 0.21 | **0.31** | **51.57** |
| **FMPlug-W** | 27.05 | 0.72 | 0.44 | **0.18** | 0.21 | 34.47 | 28.01 | 0.76 | 0.40 | 0.16 | 0.28 | 43.86 |
| **FMPlug** | **27.22** | **0.73** | **0.43** | **0.18** | 0.19 | 36.00 | **28.63** | **0.79** | **0.37** | **0.14** | 0.30 | 48.07 |

## A.5 VISUALIZATION

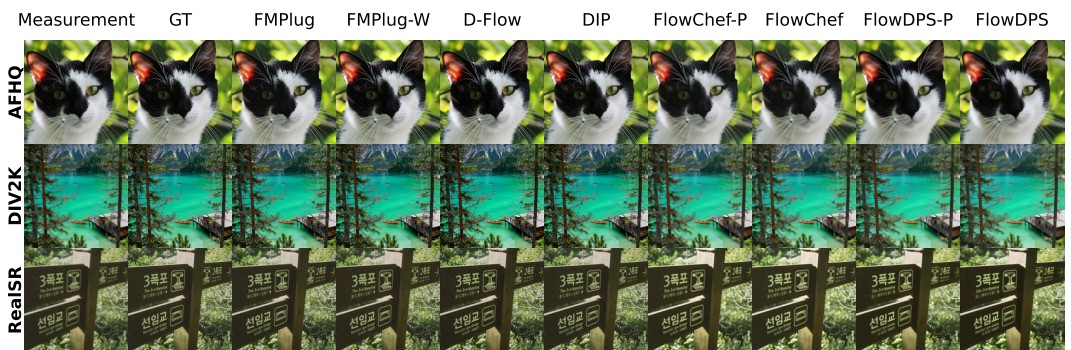

Figure 6: Qualitative comparison in super resolution $4\times$ task.

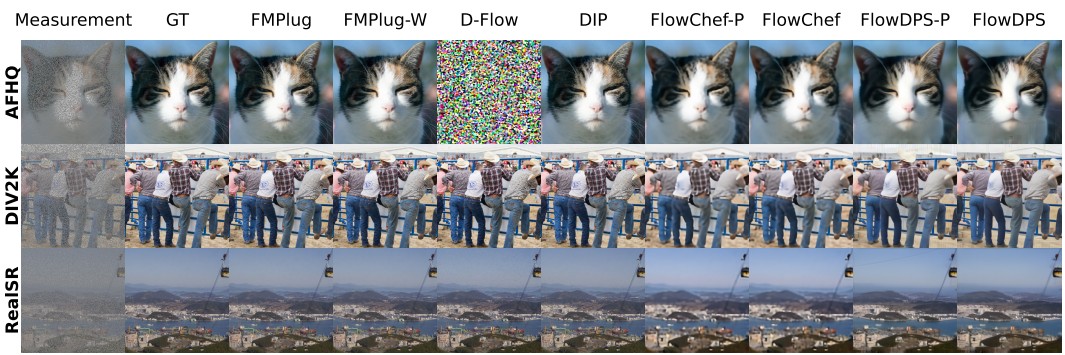

Figure 7: Qualitative comparison in Inpainting task.

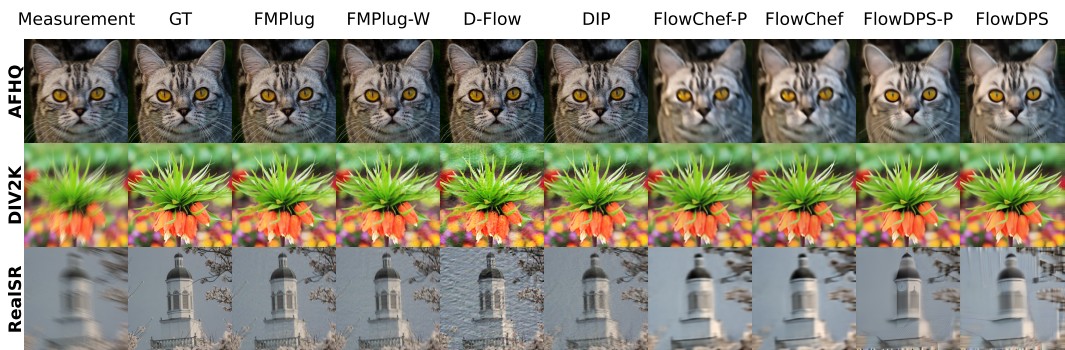

Figure 8: Qualitative comparison in motion deblur task.

