# OpenReview forum: "Saving Foundation Flow-Matching Priors for Inverse Problems"
_ICLR.cc/2026/Conference — ICLR 2026 Conference Withdrawn Submission_

### Official Review · Reviewer_9dJK · 2025-10-14

**Soundness:** 2
**Presentation:** 3
**Contribution:** 2
**Rating:** 2
**Confidence:** 4

**Summary:**

This paper proposes to use flow matching models as foundation models in inverse problems, which means that we do not retrain but employ "general" purpose FM models and hope that they work well enough on other datasets they were not necessarily trained on. The paper recalls many FM regularization strategies, most notably Flow-DPS and D-Flow, and applies them to several datasets with both domain specific and general purpose models.

**Strengths:**

The idea to use flow matching models as foundation models is timely and certainly in fashion. Furthermore the authors compared to a plethora of other methods. The improvements proposed for D-Flow seem somewhat hacky but seem to have a huge effect on performance.

**Weaknesses:**

1) Choice of foundational models: The focus on D-Flow and Flow DPS is unjustified. First of all, D-flow has the clear disadvantage that one needs to backprop through the ode solution process, which results in them using 5 (ish) Euler steps. Furthermore, they need several (pretty cool) hacks to make it work: they use a very specific optimizer and need to regularize the z so that it lands in the typical set of the gaussian. While all of this is not super bad, they are outperformed by newer methods like OT-ODE and PnP-Flow, which are much cheaper and less tune intensive. So this choice needs to be motivated better.

2) Confusion about the tables: Further, I am super confused by the results in table 1. It seems like Flow DPS is outperformed by D-Flow at around 6 PSNR, which is probably as much as youd get with a TV regularizer. This is somehow very hard to believe for me. For instance in the PnP-Flow paper D-Flow is outperformed and Flow-DPS is in my understanding a lot like PnP-Flow (although Flow DPS take latent/text models too). If flow DPS is really that much outperformed then you should look into PnP-Flow/OT-ODE/... and benchmark those.

Then looking at table 4, I think this is also some apples/oranges comparison. Although the comparison to domain specific models is nice, why not use all models data set specific and in a foundation fashion? From looking at the results, D-Flow (which your FM plug is based on) is pretty badly outperformed. Then, some metrics FM plug is pretty outperformed MUSIQ and CLIPIQ (which I guess is okay but needs some comment?), on other ones good models perform bad, that I really cannot paint a complete story from looking at the metrics. This seems more like a question one whether one prefers smooth (PSNR) images or realistic looking ones, which is something that should be concluded from these metrics.

Also in table 5, RedDiff seems to be super strong. Since diffusion models can simply be converted to flow matching models (at least for gaussian prior), it might be worthwhile to include a comparison with RedDiff as a foundation model? In figure 5 redDiff does not have an image for comparison...

3) What about your changes to a DS model?: Another point is that the improvements you make to D-Flow could also be used in a domain specific fashion, right? So it would be interesting what the gap is there.

**Questions:**

Can you outline the reasons why some of the models are not used as foundation models and others are? It would be much clearer to just take all of them as foundation models to try to benchmark them. Also including textual prompts should be straight forward for most of them, if this is the limiting factor.

What are the compute times for the different methods? Why did you focus on Flow dps and D-Flow? (flow dps does not seem to be published yet).

For instance in Fig. 7 D-Flow diverges. Did you notice similar behavior for your proposed method?

-----------------------------------------------------------------------------------------------------------------------------------------------------------------------------------
OVERALL RATING JUSTIFICATION:


The paper attempts to solve a very important and timely challenge. It does so in a very confusing way. As of now, it is super unclear why you chose these two methods to investigate foundation models. A much fairer story would be to take 5 well performing FM models and then to benchmark them as foudation models, occasionally supplying domain specific comparisons (although I am not sure how fair they are, since the DS models are much smaller, often less tuned...). If the authors can explain why they did benchmark these specific models and exclude other ones (that seem to perform well in DS setting), then I am willing to raise my rating. Does it have to do with the few shot setting?


-------------------------------------------------------------------------------------------------------------------------------------------------------------------------------------
ACTIONABLE FEEDBACK:

1) Include all main methods as foundation models on at least some of the experiments.

2) Explain the disagreement between the different metrics.

3) Explain why Flow DPS is so bad at deblurring, and why this specific example was chosen in your table 1, since it does not seem to be the normal case.

4) Apply your imrpovements also to the DS D-Flow!

5) Report computing times and memory resources.

---

### Official Review · Reviewer_R5RY · 2025-10-27

**Soundness:** 3
**Presentation:** 2
**Contribution:** 2
**Rating:** 2
**Confidence:** 4

**Summary:**

The authors introduce a
plug-in framework for foundation FM, called FMPlug towards reusable priors for soving inverse problems.
Based on the problematic in the warm-start strategy in D-flow, the main idea consists in a time-dependent warm up strategy.
The method apperas to give good (in particular sharp) results in inverse problems in image resoration and a linear inverse scattering problem when a few shot approach is added.

**Strengths:**

The objective of the paper adding problem specific guidance to foundation flow matching
is interesting as well as the few-shot setting considered in Subsection 3.3. Indeed many current flow-matching based methods for solving inverse problems rely on a pre-trained model which is domain specific (e.g. only faces or  animals).

**Weaknesses:**

Overall the contribution is incremental (in contrast to the title).
The authors merely modify D-Flow, which itself is a rather ad hoc method lacking strong theoretical grounding.
Their modifications to D-Flow are completely empirical  (e.g. the time-dependent warm up strategy (3.4): it is not even clear the image $\alpha_t y + \beta_t z$ is somewhere on a trajectory).
The numerical results look good, but visually not better than D-flow.
Further, there is neither supplementary material nor program code so that I could verify the results.

**Questions:**

- Sometimes references appear just random to me.
For example, there is a huge amount of papers on NN pre-trained priors
and it would make more sense to cite an overview paper as, e.g.
J. Hertrich et al.: Learning Regularization Functionals for Inverse Problems: A Comparative Study
- Explain the role regularizer in (2.3) better regarding that the generative model has already learned to sample from the prior.
Here (3.4) looks more ''natural'' to me.
- In line 044 is a wrong A (bold)
- The sign $\ll$ in the title of subsubsection 2.3.1 is cumbersome.
- Sometimes it is not clear to me why parts in the introduction are bold. Is this just ChatGPT generated?
- Could the authors denote the regularizer in a different way; usually $\Omega$ is used for a domain
- The numerical results look good, but visually not better than D-flow. Could the authors point/zoom to some places, where I could see differences-
Further only Tab. 4 compares with OT-ODE, PnPFlow. Why this is not done in the other experiments, just FlowDPS appeared there.
- concerning inpainting only examples with random masks are given. In this case already classical methods (see papers of R. Chahn and M. Nikolova)
work well and I like to see an example e.g. with a box mask.
- p.4, Table 2: I don't understand what they want to show there, why "the desired
surjectivity seems to hold approximately based on our image regression test reported in Table 2."

---

### Official Review · Reviewer_vg6n · 2025-10-30

**Soundness:** 3
**Presentation:** 2
**Contribution:** 2
**Rating:** 4
**Confidence:** 3

**Summary:**

The paper starts from the following two observations:
- Today, most state-of-the-art image restoration methods leverage generative models (diffusion or flow-matching models).
- Most methods that use generative models to solve inverse problems in image restoration are usually domain-specific: they are only tested on data that come from the same distribution (in practice, the same dataset split into train and test) as the one the generative model was trained on.

The idea of the paper is to adapt these methods so that they work on various datasets when used with a single foundation model such as Stable Diffusion.
Specifically, the paper adapts an existing flow-matching-based method, called D-Flow, so that it becomes compatible with such foundation models and can generalize to unseen data distributions.

To recall, D-Flow aims to solve a variational problem, but instead of solving it directly in the image space, it operates in the source space, optimizing the source (noise) point $z_0 \sim p_0$.

In practice, the authors bring two main modifications to D-Flow:
1. **Initialisation (the warm start strategy)**: instead of initializing at $t=0$ with an interpolation between the preimage of the observation $y$ (via solving the ODE backward) and Gaussian noise, they propose to initialise at $t_0 > 0$ from an interpolation of the observation $y$ and noise $z$. The pair $(t_0, z)$ is chosen to minimize the reconstruction error.
2. **Regularisation (the sharp Gaussianity regularization)** D-Flow suggests several regularization terms, one of which penalizes a function of $||z_0||^2_2$. The authors replace this regularization by a projection step.

An additional extension is proposed for the few-shot setting: in this case, initialization no longer relies on the observation $y$, which can differ a lot from the ground-truth, but instead on a few examples from the test data.

**Strengths:**

- The general problem tackled by the paper, i.e. adapting inverse problems methods based on generative models to data unseen during training, is very interesting and challenging.
Indeed, the strong dependency of these methods on the data they have been trained on, limit a lot their use in practice.
Tackling this limitation, and extending such methods to few-shot scenarios, is both important and, as far as I know, original.

-  The motivation is clearly presented. The authors demonstrate that existing approaches, when directly combined with foundation models such as Stable Diffusion, fail to generalize and even underperform naive priors such as the Deep Image Prior baseline (which uses untrained models, with random weights).

- Experimentally, the authors demonstrate good results, with a clear gain over the standard D-Flow algorithm combined with a foundation model.

**Weaknesses:**

- My main concern is that the image restoration tasks tackled by the paper (as shown is the experiments) are tasks that could be considered *non-generative* in nature (e.g. there is no mask-based inpainting).
This may partly explain the surprisingly strong performance of Deep Image Prior (an untrained prior).
In such settings, it is unclear whether leveraging a generative model is necessary at all.
I would like to see a comparison against non-generative deep-learning-based restoration methods, such as a standard Plug-and-Play framework with a pretrained deep denoiser like DPIR (Zhang, Kai, et al. "Plug-and-play image restoration with deep denoiser prior." IEEE Transactions on Pattern Analysis and Machine Intelligence (2021)).

- A second concern is *computational efficiency*. While the paper reports the number of function evaluations (NFE) in the experiments, this metric alone does not really reflect the computational and memory cost of the approach.
D-Flow is known to be expensive, both in time and memory, due to backpropagation through the ODE solver, which is also present here.
It would be valuable to provide a quantitative comparison of runtime and memory consumption against existing methods.

- On clarity and reproducibility, it seems the code is unavailable, and some points are still unclear to me regarding what is done in practice.
Writing explicitely the final algorithm (pseudocode) would help clarifying the following questions (see Questions section).

**Questions:**

On the algorithm itself:
1. Is the warm-start strategy an initialisation ? If yes, by what it is followed ? Or is it the end of the procedure, meaning the reconstruction image is  $\mathcal G(z_{t_0}, t_0)$ where $(t_0, z_{t_0})$ is the solution of the warm star strategy  ?
2. Is the projection-based regularization applied jointly during the warm start strategy ?

**Typos and minor comments**

- In the introduction, the sentence: "On the other hand, the emergence of domain-agnostic foundation FM models, for images, obsoletes domain-specific developments; Kim et al. (2025); Patel et al. (2024); Ben-Hamu et al. (2024); Martin et al. (2025) propose such ideas." is confusing. It seems to suggest that all of these methods are domain-agnostic and leverage foundation models.
However, as far as I can tell, except for FlowDPS (Kim et al.), most of the cited approaches remain domain-specific: their experiments are conducted on test data drawn from the same distribution as their training data.

- Notations in Figure 1 are not very clear: what is $\epsilon_\theta^\star$, is it the velocity field (previously denoted as $v_\theta$) ? What is $\mathcal R$, is it the generator $\mathcal G_\theta$?

- Typo l051: citep is missing.
- Typo l385: AFHQ dataset (the A is missing)

---

### Official Review · Reviewer_NZ51 · 2025-11-01

**Soundness:** 2
**Presentation:** 1
**Contribution:** 2
**Rating:** 2
**Confidence:** 3

**Summary:**

The authors propose a way to use foundation flow-matching priors to solve inverse problems, called FMPlug. FMPlug utilizes an instance-guided warm-start strategy to help with sampling. It also involves sharp Gaussianity regularization. With their experiments, the authors find the warm-start strategy to be most helpful, with the improved Gaussianity regularization helping with finer details. The authors compare with an untrained prior (DIP), state-of-the-art foundation-FM-based methods, and a method using a domain-specific FM model. They find that their method mitigates the phenomenon of foundation FMs performing worse as priors compared to domain-specific FMs and even untrained priors like DIP.

**Strengths:**

* The authors provide rather extensive experimental results with a variety of baselines and inverse problems.

**Weaknesses:**

* The assumptions that make the warm-start strategy so effective don’t seem to be that practical. In the simple distortion case, it makes sense that the warm-start strategy would make a general foundation FM more effective because it gives more information about the true image. I don’t find the scientific IP case that compelling, either. How often is it the case that we have a small dataset of images that already look “similar” to the true image?
* I have a slight philosophical disagreement with the authors. In principle a foundation FM prior should in fact be “worse” than a domain-specific FM prior because it has less bias. FMPlug works because it imposes more bias through information about the true instance-specific image. In that sense it is not necessarily calling upon the true prior of the foundation FM.
* The writing is quite poor, with a lot of grammatical errors throughout that make it a bit difficult to read.

**Questions:**

* Perhaps the authors can address my question about why we would want foundation FM priors to be better than domain-specific ones.
* Please address my concern about the assumption of having a dataset of similar images in the case of scientific IPs.

---

### Note · Authors · 2025-11-19

I have read and agree with the venue's withdrawal policy on behalf of myself and my co-authors.